# Characterizing Early Cardiac Metabolic Programming via 30% Maternal Nutrient Reduction during Fetal Development in a Non-Human Primate Model

**DOI:** 10.3390/ijms242015192

**Published:** 2023-10-14

**Authors:** Susana P. Pereira, Mariana S. Diniz, Ludgero C. Tavares, Teresa Cunha-Oliveira, Cun Li, Laura A. Cox, Mark J. Nijland, Peter W. Nathanielsz, Paulo J. Oliveira

**Affiliations:** 1Laboratory of Metabolism and Exercise (LaMetEx), Research Centre in Physical Activity, Health and Leisure (CIAFEL), Laboratory for Integrative and Translational Research in Population Health (ITR), Faculty of Sports, University of Porto, 4200-450 Porto, Portugal; 2CNC-UC, Center for Neuroscience and Cell Biology, University of Coimbra, 3004-504 Coimbra, Portugal; mdiniz@cnc.uc.pt (M.S.D.); teresa.oliveira@cnc.uc.pt (T.C.-O.); pauloliv@ci.uc.pt (P.J.O.); 3CIBB—Centre for Innovative Biomedicine and Biotechnology, University of Coimbra, 3004-504 Coimbra, Portugal; 4Center for Pregnancy and Newborn Research, University of Texas Health Science Center at San Antonio, San Antonio, TX 78229, USA; nijland@uiwtx.edu; 5PDBEB—Ph.D. Programme in Experimental Biology and Biomedicine, Institute for Interdisciplinary Research (IIIUC), University of Coimbra, 3030-789 Coimbra, Portugal; 6CIVG—Vasco da Gama Research Center, University School Vasco da Gama—EUVG, 3020-210 Coimbra, Portugal; 7Texas Pregnancy & Life-Course Health Research Center, Department of Animal Science, University of Wyoming, Laramie, WY 82071, USA; cunli1@gmail.com; 8Center for Precision Medicine, Wake Forest University Health Sciences, Winston-Salem, NC 27157, USA; laurcox@wakehealth.edu (L.A.C.); peter.nathanielsz@yahoo.com (P.W.N.); 9Section on Molecular Medicine, Department of Internal Medicine, Wake Forest University School of Medicine, Winston-Salem, NC 27157, USA; 10Southwest National Primate Research Center, Texas Biomedical Research Institute, San Antonio, TX 78227, USA; 11Section on Comparative Medicine, Department of Pathology, Wake Forest University School of Medicine, Winston-Salem, NC 27157, USA

**Keywords:** fetal plasticity, DOHaD development origins of health and diseases, nutrition during pregnancy, macronutrients, fetal metabolism, cardiometabolic programming

## Abstract

Intra-uterine growth restriction (IUGR) is a common cause of fetal/neonatal morbidity and mortality and is associated with increased offspring predisposition for cardiovascular disease (CVD) development. Mitochondria are essential organelles in maintaining cardiac function, and thus, fetal cardiac mitochondria could be responsive to the IUGR environment. In this study, we investigated whether in utero fetal cardiac mitochondrial programming can be detectable in an early stage of IUGR pregnancy. Using a well-established nonhuman IUGR primate model, we induced IUGR by reducing by 30% the maternal diet (MNR), both in males (MNR-M) and in female (MNR-F) fetuses. Fetal cardiac left ventricle (LV) tissue and blood were collected at 90 days of gestation (0.5 gestation, 0.5 G). Blood biochemical parameters were determined and heart LV mitochondrial biology assessed. MNR fetus biochemical blood parameters confirm an early fetal response to MNR. In addition, we show that in utero cardiac mitochondrial MNR adaptations are already detectable at this early stage, in a sex-divergent way. MNR induced alterations in the cardiac gene expression of oxidative phosphorylation (OXPHOS) subunits (mostly for complex-I, III, and ATP synthase), along with increased protein content for complex-I, -III, and -IV subunits only for MNR-M in comparison with male controls, highlight the fetal cardiac sex-divergent response to MNR. At this fetal stage, no major alterations were detected in mitochondrial DNA copy number nor markers for oxidative stress. This study shows that in 90-day nonhuman primate fetuses, a 30% decrease in maternal nutrition generated early in utero adaptations in fetal blood biochemical parameters and sex-specific alterations in cardiac left ventricle gene and protein expression profiles, affecting predominantly OXPHOS subunits. Since the OXPHOS system is determinant for energy production in mitochondria, our findings suggest that these early IUGR-induced mitochondrial adaptations play a role in offspring’s mitochondrial dysfunction and can increase predisposition to CVD in a sex-specific way.

## 1. Introduction

Cardiovascular disease (CVD) incidence is increasing worldwide at an alarming rate, especially among the young adult population [1,2]. In comparison with >50-year-old adults, the incidence of CVD for younger adults for the same period is either similar or has increased [1]. CVD risk is sex-specific, since men present an increased risk to develop CVD at an earlier stage in life than women [3].

It is now well accepted that maternal nutrition influences the intrauterine environment and, consequently, the offspring’s short- and long-term health [4,5]. Maternal nutrient restriction is a common cause of intrauterine growth restriction (IUGR) [6], in which the fetus might not reach its full growth potential [7]. Multiple studies have provided evidence that fetuses experiencing nutrient deprivation during development exhibit a phenotype characterized by being born small for gestational age (SGA) [8]. These infants often display cardiovascular abnormalities in comparison to infants who have achieved appropriate size for gestational age. Additionally, these individuals commonly exhibit impaired heart systolic and diastolic functions [9]. Animal studies support progeny’s cardiac remodeling in IUGR conditions characterized by reduced cardiomyocyte maturation [10], and increased apoptotic rates [9].

Using a well-established nonhuman IUGR primate model of moderate maternal nutrient reduction (MNR), which consisted of a 30% nutrient reduction in comparison with control mothers, and that shares a high level of gene homology with humans (94%) [11], we previously showed that term IUGR offspring (165 days old—0.9 gestation, 0.9 G) [12] exhibited disrupted cardiac mitochondrial fitness, with a two-fold increase in fetal cardiac left ventricle (LV) mitochondrial DNA (mtDNA), increased transcripts levels for several respiratory chain subunits and increased abundance for the mitochondrial proteins NADH dehydrogenase (ubiquinone) 1 beta subcomplex, 8 (NDUFB8), ubiquinol-cytochrome c reductase core protein I (UQCRC1), and cytochrome c and adenosine triphosphate (ATP) synthase. However, IUGR fetal cardiac mitochondria displayed significantly decreased complex-I and -II/III activities, possibly contributing to the 73% decreased ATP content and increased lipid peroxidation. The impairment of oxidative phosphorylation (OXPHOS) due to mitochondrial oxidative damage can contribute to an exacerbation of reactive oxygen species (ROS) production and oxidative disturbances [13]. This creates a detrimental feedback loop, where increased ROS production further damages mitochondria and perpetuates oxidative stress [14]. MNR fetal left ventricle (LV) also showed mitochondrial dysmorphology with sparse and disarranged cristae. Furthermore, our findings demonstrated that MNR induced sex-specific adaptations in cardiac mitochondrial biology among term fetuses in this animal model [12]. This suggests that an early-impaired cardiac mitochondrial function could predispose MNR offspring to an increased CVD risk later in life [12]. Right before birth, term MNR fetuses presented cardiac structural and functional alterations (i.e., extracellular fibrosis, miRNA expression levels, lipid metabolism [12]), which could be indicative of MNR-induced early cardiac adverse adaptations in this animal model [12]. After birth, MNR offspring presented 11% less weight than control offspring [15]. In fact, in this non-human primate (NHP) model, 3.5-year-old MNR offspring fed a control diet postnatally presented cardiac abnormalities, such as myocardial remodeling, impaired physiological function in both ventricles and altered diastolic and systolic functions [12,16]. During the young and adult phases of the offspring, mitochondrial dysfunction can become accentuated, leading to an increased occurrence of mitophagy (selective degradation of damaged mitochondria [17]) and mitochondrial membrane permeabilization [18]. This process enables the release of mitochondrial DNA (mtDNA) into the cytosol [19], which, in turn, activates inflammatory [20] and apoptotic pathways [21]. In utero programming of mitochondrial function likely contributes to the documented developmental programming of adult cardiac dysfunction, indicating a programmed mitochondrial inability to deliver sufficient energy to cardiac tissues as a chronic mechanism for later-life heart failure [22]. However, little is known about the in utero fetal cardiac mitochondrial nutritional plasticity and how early these metabolic adaptations can be detected. This information is relevant to understanding the etiology of several mitochondrial-related human diseases.

Despite the glycolytic dependence of fetal cardiomyocytes [23] and that during early embryonic stages, oxygen availability is reduced in comparison with postnatal stages [24] (highly aggravated by IUGR, which has been suggested to induce a hypoxic environment) and mitochondrial OXPHOS system activity is decreased [25], mitochondria play a crucial role supporting early-stage fetal growth [26] and cardiac development [27]. For instance, mitochondrial oxidative metabolism is essential for cardiac differentiation from embryonic stem cells (i.e., cardiac specification and contraction ability) [28,29], which starts in the first few weeks of pregnancy both in humans and in NHP [30,31]. Currently, it still remains unknown at which fetal stage MNR-induced cardiac mitochondrial-related alterations first occur. Thus, this study aimed to determine whether a 30% global MNR induces detectable in utero adaptations on fetal baboon left cardiac ventricular mitochondrial transcripts and protein content at the mid-gestation period (0.5 gestation, 0.5 G) and whether these are sex-specific. We hypothesized that maternal malnutrition during pregnancy impairs in utero fetal cardiac mitochondrial regulation at an early stage during fetal development, imprinting cardiac adaptations during the prenatal period that may impact postnatal mitochondrial cardiac function, contributing to offspring’s increased risk of CVD later in life.

## 2. Results

### 2.1. MNR-Induced Maternal and Fetal Morphometrics Alterations Accompained by Fetal Blood Biochemistry Changes

Non-pregnant outbred female baboons (*Papio* spp.) with similar morphometrics, including pre-pregnancy body weight (13.9 ± 0.5 vs. 14.26 ± 0.6), were randomly assigned to the Control (C) or MNR group. After diet treatment, maternal weight variation differed between groups, being increased for MNR mothers vs. C mothers (−0.11 ± 0.12 vs. −0.72 ± 0.15 kg; *p* = 0.008) (Table 1).

As for fetal morphometrics (Table 1) and blood parameters (Table 2), fetal weight did not vary between groups at 0.5 G (101.1 ± 3.2 vs. 101.2 ± 3.0 g). The heart weight to body weight (HW/BW) ratio was increased for C-Females (F) in comparison with C-M (11.1 ± 3.7 vs. 5.5 ± 0.6; *p* = 0.047). For the same group, C-F fetal body mass index was decreased vs. C-M (3.1 ± 0.1 vs. 3.6 ± 0.2; *p* = 0.045). Fetal blood biochemical analysis revealed that fetal glucose circulating levels were decreased for MNR-F vs. C-F (34.7 ± 2.7 vs. 52.2 ± 3.4; *p* = 0.036). This was not observed for maternal blood glucose concentration. Interestingly, fetal blood urea nitrogen (BUN) was decreased for MNR vs. C (8.0 ± 0.5 vs. 11.0 ± 0.6; *p* = 0.004), being more evident for MNR-F vs. C-F (8.7 ± 0.7 vs. 11.6 ± 0.6; *p* = 0.016), and for C-F vs. C-M (11.6 ± 0.6 vs. 9.5 ± 0.5; *p* = 0.033). Fetal BUN/creatinine ratio was decreased for MNR vs. C (10.0 ± 1.0 vs. 14.0 ± 1.0; *p* = 0.035) (Table 2).

The cluster heatmap analysis identified a reasonable separation between the experimental groups’ amino acid blood levels, resulting in well-formed clusters as seen in the heatmap (Figure 1). In general, MNR amino acid blood levels were increased in comparison with the C group. The differences between both groups were more pronounced in alanine, glycine, and taurine, all of which were increased in MNR vs. C, in agreement with published data for this model [6].

### 2.2. No Alterations Were Detected in Cardiac Mitochondrial DNA Copy Number

To evaluate MNR-induced effects on mitochondrial biology, mtDNA copy number was determined via qRT-PCR, which was calculated by the ratio between the absolute amount of mitochondrial gene *ND1* versus the absolute amount of the *B2M* nuclear gene for each sample. No statistically significant differences were detected for LV cardiac mtDNA copy number at 0.5 G between C and MNR offspring (Appendix A).

### 2.3. MNR-Induced Mitochondrial Transcriptional Alteration in the Fetal Cardiac Left Ventricle at 0.5 Gestation

Human Mitochondrial Energy Metabolism and the Human Mitochondria Pathway Arrays were used to evaluate MNR-induced in utero fetal cardiac RNA transcriptional changes at 0.5 G. The expression levels of all of the evaluated genes are summarized in the heatmap and the scatter plot in Figure 2A and Table 3. The clustering analysis shows a clear separation between MNR and C groups. The fold-regulation between MNR and Control gene expression levels (Figure 2A–C, and Table 3) showed that transcripts for *HSPD1* (fold-regulation: −1.6229, *p* = 0.0097), *BCS1L* (fold-regulation: −1.3446, *p* = 0.0089, *LHPP* (fold-regulation: −1.4705, *p* = 0.0026, *ATP4A* (fold-regulation: −1.5006, *p* = 0.0199) were downregulated, whereas the genes *NDUFB6* (fold-regulation: 1.2714, *p* = 0.0474), *NDUFB3* (fold-regulation: 1.2952, *p* = 0.0125), *UQCRC2* (fold-regulation: 1.3692, *p* = 0.0274), *ATP5O* (fold-regulation: 1.1510, *p* = 0.0373), and *ATP5A1* (fold-regulation: 2.2; *p* = 0.0089) were upregulated. Moreover, two genes had a tendency to be downregulated (*CDKN2A*, fold-regulation: *p* = 0.0558; *NDUFB7* (*p* = 0.0587)), and five genes had a tendency to be upregulated (*COX6C* (*p* = 0.0714), *BAK1* (*p* = 0.0856), *ATP5J* (*p* = 0.0588), *TIMM10* (*p* = 0.0762), *ATP5F1* (*p* = 0.0595)).

Gene expression variations are sex-specific (Figure 3 and Table 3). In common, *ATP5A1* was upregulated and *LHPP* was downregulated in both MNR-M (fold-regulation: 2.3, *p* = 0.0004; fold-regulation: −1.2608; *p* = 0.0369) and MNR-F (fold-regulation: 2.2; *p* = 0.0419; fold-regulation: −1.7836; *p* = 0.0142) in comparison with the control of the respective sex. For MNR-M (Figure 3B and Table 3), along with *ATP5A1*, four more transcripts were upregulated in this group: *UQCRC2* (fold-regulation: 1.2991, *p* = 0.0298), *NDUFV3* (fold-regulation: 1.4313; *p* = 0.0382), *NDUFB10* (fold-regulation: 1.8811; *p* = 0.0035, *TOMM20* (fold-regulation: 1.1394; *p* = 0.0481), and *COX10* was tendentially upregulated (*p* = 0.0827). For the same group, along with *LHPP*, the transcripts for *UCP2* (fold-regulation: −1.6445, *p* = 0.0155), *SLC25A31* (fold-regulation: −1.8892; *p* = 0.0416), *IMMP1L* (fold-regulation: −1.4269; *p* = 0.0070), *TIMM50* (fold-regulation: −1.6130; *p* = 0.0319), and *TIMM22* (fold-regulation: −1.8063; *p* = 0.0223) were downregulated in comparison with C-M, and another seven transcripts were tendentially downregulated, *SLC25A2* (*p* = 0.0677), *PMAIP1* (*p* = 0.0770), *CPT2* (*p* = 0.0958), *TP53* (*p* = 0.0577), *SOD2* (*p* = 0.0620), *BID* (*p* = 0.0759), and *BBC3* (*p* = 0.0848).

In comparison with C-F (Figure 3C and Table 3), only 1 transcript besides *ATP5A1* was upregulated for MNR-F: *NDUFB3* (fold-regulation: 1.4818, *p* = 0.0207). For the same group, 10 genes were downregulated along with *LHPP*: *HSPD1* (fold-regulation: −2.0; *p* = 0.0108), *CYC1* (fold-regulation: −2.7; *p* = 0.0039), *UQCRC1* (fold-regulation: 1.5758, *p* = 0.0089), *SDHB* (fold-regulation: −1.2856, *p* = 0.0066), *NDUFA10* (fold-regulation: −1.2561, *p* = 0.0236), *COX4I2* (fold-regulation: −1.5859, *p* = 0.0211), *NDUFS2* (fold-regulation: −1.5158, *p* = 0.0022), *BCS1L* (fold-regulation: −1.7991, *p* = 0.0015), *ATP4A* (fold-regulation: −1.9599, *p* = 0.0182), along with other four tendentially downregulated transcripts, *COX7A2L* (*p* = 0.0671), *NDUFA3* (*p* = 0.0942), and two tendentially upregulated, *SLC25A5* (*p* = 0.0868), NDUFB6 (*p* = 0.0974).

When comparing the fold-regulation of gene expression in MNR-M vs. MNR-F, Figure 4A and Table 3 (section MNR-M vs. MNR-F) show that the upregulated transcripts for MNR-M were *ATP5A1*, *ATP5G3*, *HSPD90AA1*, *IMMP1L*, *SLC25A2*, *SLC25A31*, *TIMM22*, *TOMM70A*, and *UCP2*, with *ATP5C1*, *COX6A2*, *NDUFV1*, *CPT2*, *DNM1L*, *MFN1*, *OPA1*, and *TP53*, being tendentially upregulated, and for MNR-F, the upregulated genes were *COX4I2*, *NDUFV3*, *CYC1*, *MIPEP*, *TOMM40*, *ATP4A*, *UQCRC1*, *UQCRFS1*, *SDHB*, *SDHD*, *NDUFA11*, *NDUFC1*, *NDUFS2*, *NDUFS3*, *TOMM20*, *BCS1L*, *NDUFA1*, *NDUFA10*, *COX412* with *ATP5B*, *LHPP*, *HSPD1*, *SLC25A19*, *TIMM17A*, *TIMM44*, and *TOMM22*, being tendentially upregulated.

Overall, fold-regulation patterns are patented in Figure 4B displaying the sex-specific regulation patterns dictated by MNR in the cardiac LV tissue. When comparing with same-sex control, MNR-induced transcripts fold-regulation is distinct for most analyzed genes, with several genes presenting tendential opposite patterns of expression depending on the sex of the offspring. For example, transcripts that tended to be upregulated in MNR-M vs. C-M were *NDUFB10*, *NDUFA3*, *NDUFV3*, *NDUFS2*, *SDHB*, *UQCRC1*, *COX7A2L*, *COX4I2*, and *CYC1* for MNR-F vs. C-F; these transcripts tended to be downregulated. On the contrary, the group of transcripts that tended to be downregulated for MNR-M vs. C-M was as follows: *BBC3*, *BID*, *CPT2*, *PMAIP1*, *SLC25A2*, *SLC25A31*, *SOD2*, *TIMM22*, *TIMM50*, and *TP53*; these were upregulated for MNR-F vs. C-F. Nevertheless, a few genes show similar expression behaviors for both sexes (*NDUFB3*, *NDUFB6*, *UQCRC2*, *COX10*, *ATP4A*, *ATP5A1*, *LHPP*, *UCP2*, *BCS1L*, *HSPD1*, *SLC25A5*).

### 2.4. MNR-Induced Mitochondrial Protein Modulation in the Fetal Cardiac Left Ventricle at 0.5 Gestation

The transcriptional changes were complemented with protein content analysis performed via Western blot (WB). Protein indicators of mitochondrial mass, dynamics, oxidative stress, and OXPHOS subunit proteins were semi-quantified (Figure 5). Protein amounts for citrate synthase were decreased for MNR vs. C (0.69 ± 0.16 vs. 0.9 ± 0.15) and protein amounts for complex IV subunit COX6C were increased for MNR vs. C (1.15 ± 0.14 vs. 1.0 ± 0.1). In the hearts of male fetuses significant alterations were identified in the protein amounts for OXPHOS subunits complex I NDUFB8 (1.7 ± 0.27 vs. 1.0 ± 0.22; *p* = 0.0485), complex III UQCRC2 (1.8 ± 0.3 vs. 1.0 ± 0.26, *p* = 0.0330), and complex IV COXII (1.6 ± 0.11 vs. 1.0 ± 0.22; *p* = 0.0245) subunits, in which OXPHOS protein content overall increased for MNR-M vs. C-M. Additionally, in MNR-F fetuses there was an increase in LV protein content for the complex IV COX6C subunit protein (MNR-F vs. C-F, 1.3 ± 0.03 vs. 1.0 ± 0.07, *p* = 0.0063) (Figure 5).

Cardiac mitochondrial sexual dimorphism was well evidenced in the protein levels between the control groups (Figure 5, C-F vs. C-M) for the protein NDUFB8 (1.9 ± 0.49 vs. 1.0 ± 0.22, *p* = 0.0285), complex III UQCRC2 (1.9 ± 0.33 vs. 1.0 ± 0.26, *p* = 0.0225), complex IV COXII (1.7 ± 0.1 vs. 1.0 ± 0.2; *p* = 0.0068), and ATP synthase ATP5A1 subunit (1.8 ± 0.3 vs. 1.0 ± 0.20; *p* = 0.0393). Mitochondrial biogenesis and mass regulator mitochondrial transcription factor A (TFAM) (Figure 5G; 1.4 ± 0.12 vs. 1.0 ± 0.14, *p* = 0.05), as well as the translocase of the outer mitochondrial membrane 20 (TOM20) (Figure 5D; 1.6 ± 0.04 vs. 1.0 ± 0.07, *p* = 0.0004), respectively, were also increased for C-F vs. C-M. Comparing both MNR groups, the sexual dimorphism regarding mitochondrial protein amount was attenuated with only OXPHOS complex IV COX6C subunit (1.3 ± 0.03 vs. 1.0 ± 0.06, *p* = 0.0034) and TOM20 (1.7 ± 0.10 vs. 1.2 ± 0.11; *p* = 0.0007) being increased for MNR-F vs. MNR-M.

## 3. Discussion

During pregnancy, the mother’s nutritional intake plays a vital role in supporting fetal growth and development [32,33,34]. Inadequate maternal nutrition, such as limiting the availability of essential fetal building blocks, can have detrimental effects on fetal growth [35,36] leading to IUGR and the development of SGA. SGA refers to infants who have a birth weight below the 10th percentile for their gestational age [37,38]. MNR is a recognized risk factor for IUGR and SGA [39,40]. However, it is not the sole cause, as other factors, such as maternal health conditions (e.g., hypertension, diabetes), smoking, drug use, and genetic factors, can also contribute to these conditions [41,42].

IUGR is a major obstetric condition that prompts the fetus and, later on, the offspring, for increased CVD risk [43]. We have previously reported that a 30% MNR has an adverse impact on cardiac left ventricle (LV) mitochondria in a sex-dependent way in term NHP fetuses (0.9 of gestation) [12]. Given the crucial role of mitochondria for cardiac differentiation and contraction in the first trimester of gestation [44], the objective of this study was to investigate whether the in utero programming of fetal cardiac mitochondria could be discernible during the early stage of pregnancy in the NHP animal model (90 days of gestation, 0.5 G). We aimed to assess the effects of a 30% maternal nutrient restriction (MNR) on the mitochondrial biology of the fetal heart during mid-gestation, using the same NHP animal model. In this study, the analysis of the cardiac LV of fetuses at 0.5 G showed that MNR already induced cardiac mitochondrial biology alterations during fetal development detectable at this time point. Thus, we here provide significant clues about MNR-induced early modulation of cardiac LV mitochondrial biology, allowing a time-dependent characterization and comprehension of cardiac metabolic adaptations in MNR fetuses that programs for increased CVD risk [16].

In humans, SGA fetuses are usually hypoglycemic [45]. Our study corroborates these findings with 0.5 NHP MNR fetuses showing decreased levels of glucose in comparison with C fetuses. Hypoglycemia during fetal development adversely impacts cardiogenesis [46,47], the process of cardiomyocyte differentiation and maturation, since it mainly relies on glycolysis and lactate production until the postnatal period [48]. In the postnatal stage, fuel for cardiac metabolism shifts to primarily fatty-acid oxidation which is dependent on cardiac mitochondria [49]. This metabolic shift is necessary to induce cardiomyocyte proliferation [50]. Despite not being the primary source of energy in the cardiomyocytes during fetal development, mitochondria play an essential role in fetal cardiac development [51]. Mitochondria are important organelles for ventricular morphogenesis [51], through the regulation of apoptotic mechanisms [52], as well as in the regulation of cardiac differentiation markers, through mitochondrial fusion [53], and by the production of ROS that can act as signaling molecules [51]. Hence, impaired mitochondrial function during the early stages of development may lead to detrimental adaptations in the developing heart, giving rise to a variety of anomalies in cardiac growth and development. We here show, for the first time, that 30% MNR induces mitochondrial-related transcriptional and protein abundance levels alterations as early as the mid-gestation period (0.5 G) in NHP fetuses cardiac LV, which may become relevant to understanding the mechanisms by which MNR affects cardiac function postnatally.

Cardiac gene expression analysis of mitochondria-related genes revealed that MNR-induced gene downregulation involves genes that encode for a phosphatase enzyme (*LHPP*) and proteins involved in macromolecular assembly (*HSPD1* and *BCS1L*). It is worth mentioning that *BCS1L* plays an important role in MRC-complex III assembly [54]. Interestingly, the downregulation of *BCS1L* and OXPHOS impairment has been proposed as a mechanistic link in prostate cancer-related fatigue development; however, more studies are needed to fully understand if this link between *BCS1L* gene expression and complexes activities is verified for other conditions. Nevertheless, transcripts for *UQCRC2*, a gene encoding for complex-III subunit, was upregulated for MNR fetuses, along with other OXPHOS complex subunits: complex-I (*NDUFB6*, *NDUFB3*), -IV (*COX6C*), and ATP synthase (*ATP5A1*, *ATP5F1*, *ATP5J*, *ATP5O*). This represents more than half of the measured genes whose expression was altered by MNR in the cardiac LV. In addition, and highlighting MNR-induced mitochondrial alterations, the protein amount of citrate synthase, a mitochondrial matrix enzyme commonly used as a mitochondrial marker [55], was decreased for MNR mid-gestation fetuses, suggesting a decreased mitochondrial number. Most transcripts for OXPHOS subunit genes were upregulated, and the same was observed in our previous study for cardiac tissue from term fetuses (0.9 G) [12]. In both mid-gestation and term fetuses, transcripts for complex I subunits *NDUFB6*, and ATP synthase *ATP5A1* were upregulated. These transcript alterations did not result in many protein content alterations for the same subunits, in both fetal time points (with the exception of COX6C for MNR mid-gestation fetuses), possibly due to the dynamic processes between the genome and the protein levels, which include transcription, splicing, and translation [12]. In spite of this, in term fetuses, MRC complex-I, -II/III, and -IV activities were altered [12]. This was also observed in a diet-induced animal model of IUGR in adult mice offspring (14 weeks old). Protein content for OXPHOS subunits was unaltered by IUGR; however, mitochondrial respiration was decreased in IUGR-offspring cardiac LV muscle [56]. Therefore, to comprehensively assess the true impact of MNR on OXPHOS and its subsequent effects on cardiac mitochondria, it is essential to measure either respiratory rates or the activity of OXPHOS. However, conducting such studies can be challenging and often impractical due to limited sample availability and the significant amount of biological material required. The impact of fetal sex on mitochondrial metabolism-related gene expression and protein content was evident. The fold-regulation of MNR-M vs. MNR-F reveals a sex-specific pattern of gene expression induced by MNR. For MNR-F, the majority of upregulated transcripts in comparison with the respective C include those encoding for OXPHOS subunits, while for MNR-M, this was more evident for transcripts for other mitochondrial proteins that are involved in ornithine transport (*SLC25A2*), mitochondrial ADP/ATP exchange (*SLC25A31*), uncoupling of mitochondrial oxidative phosphorylation (*UCP2*), molecular folding (*HSP90AA1*), removal of the mitochondrial targeting pre-sequence of nuclear-encoded proteins (*IMMP1L*), and protein translocation (*TIMM22*, *TOMM70A*). Interestingly, both complex III-core 2 subunit gene expression (*UQCRC2*) and protein content were increased for MNR-M vs. C-M. Moreover, complex-I (NDUFB8) and -IV (COXII) subunit protein contents were increased for MNR-M vs. C-M, whereas no differences were detected in MNR-F. Thus, MNR seems to impact more severely male fetuses’ OXPHOS protein content, whereas for female offspring, these alterations may become more pronounced at a later developmental stage or even at a postnatal stage, since we did not detect differences in these subunits’ protein content for term female fetuses in our previous study [12]. These sex-specific differences may be explained by, e.g., epigenetic regulation, or by the action of sex-steroid hormones, given that estrogen plays an active role in protein modification, gene regulation, and cellular process modulation [57]. Nevertheless, further studies are needed to understand how sex influences in utero cardiac programming and adult offspring disease risk.

Another implication of the in utero MNR-induced OXPHOS alterations may be the impact in the production of ROS [4]. Our results showed that during mid-gestation, oxidative stress markers’ protein content was not altered, whereas we detected MNR-induced increased levels of MDA in term-fetuses [12], indicating that oxidative damage may only become evident during the second half of pregnancy in this NHP MNR model. This can relate to the unaltered mtDNA copy number, suggesting that perhaps mitochondrial mass and biogenesis are still preserved since lack of significant oxidative damage occurring at this fetal stage. Indeed, in a rabbit animal model of IUGR, in the LVs of 30-day-old fetuses (corresponding to late gestation period), increased expression of genes that modulate OXPHOS, including cardiac mitochondrial respiratory chain complex I, NADH dehydrogenase activity was detected [58]. These alterations were accompanied by decreased cardiac enzymatic activities of complex-II, -IV, and -II + III [59]. Nevertheless, these mitochondrial respiratory chain adaptations did not produce alterations in cardiac cellular ATP levels, nor in the antioxidant enzyme SOD2, nor in mitochondrial copy number [58], in accordance with our findings. Guitart-Mampel et al. suggest that the preservation of ATP levels and SOD2 protein expression levels and activity can be attributable to increased levels of Sirtuin 3 [59], acting as a compensatory mechanism, due to its action on promoting mitochondrial energy production, on the inhibition of oxidative stress, and on autophagy regulation [60]. We recognize the possibility of Sirtuin 3 acting as a compensatory mechanism in the hearts of our animal model as well, however, it must be taken into consideration that in Guitart et. al.’s study, IUGR was achieved surgically and the animal models are very distinct from each other (i.e., gestation period, diet, litter size). Highlighting this, in our animal model, term fetuses display signs of oxidative damage [12]. To our knowledge, no other study has explored Sirtuin-3 levels in the context of IUGR fetal programming of CVD, and thus, no major conclusions can be drawn.

Cardiomyocyte proliferation in the neonatal period depends mainly on oxidative metabolism [51,61]. Even though we show here that some MNR-induced alterations begin to be detectable at mid-gestation in cardiac mitochondria, these become more pronounced at late gestation, e.g., increased mtDNA content and oxidative stress in comparison with C [12]. We must consider that at 0.5 G, mitochondria are smaller and less mature [50]. Thus, it is possible that MNR-induced adaptations in fetal cardiac mitochondrial function become significant enough to impair cardiac function only at a stage where cardiomyocytes mainly rely on mitochondrial respiration as a fuel source. However, we cannot underestimate the impact of mitochondrial dysfunction and signaling dysregulation in driving proper cardiomyocyte maturation, cardiac differentiation, and heart organogenesis that may only be perceivable in advanced postnatal life stages [49,62,63]. To our knowledge, our study is the only one reporting IUGR-induced mitochondrial-related genes’ expression alterations in fetal hearts at this stage in development, and thus, we can only raise hypotheses. It has been suggested that maternal nutrient reduction leads to a hypoxic environment in fetal organs [64,65]. The literature has suggested that hypoxia alters the expression of transcription factors involved in the replication of mtDNA-encoded genes [66], which include genes from the subunits of the mitochondrial respiratory chain complexes. On top of that, it is widely accepted that hypoxia activates an oxygen-sensing transcription factor, the hypoxia-inducible factor 1α (HIF-1α) [67], which is responsible for modulating the expression of microRNAs that regulate MRC complex subunits’ assembly proteins’ expression [67], highlighting the potential role of epigenetic remodeling. In addition, because oxygen is the final electron acceptor of the mitochondrial electron transport chain (ETC), a lack of oxygen availability affects ETC function, resulting in an imbalance between oxygen and electron flow, leading to an overproduction of ROS [67], which can affect the expression of genes that encode for ETC complex subunits [68]. This is less likely to occur in an early fetal stage because, in this study, we did not find any significant alterations related to markers of oxidative stress, but this can occur in later stages since these alterations were verified in the hearts of 165-day-old fetuses [12].The initial changes that we here report, sustained throughout an individual’s lifetime, can potentially lead to a progressive disruption of cardiac metabolic functions. In spite of that, the sex-specific response is already clear at this stage, and the different patterns of gene expression according to fetal sex are significant, highlighting the need to explore CVD programming according to fetal offspring sex, highlighting the need for effective sex-specific treatment in disease management and mitigation.

## 4. Materials and Methods

### 4.1. Animal Care and In Vivo Procedures

#### 4.1.1. Animal Care and Maintenance

The Animal Care and Use Committees of the Texas Biomedical Research Institute and the University of Texas Health Science Center at San Antonio, TX (no. 1134PC) approved all animal procedures, including pain relief. These were conducted in the Association for Assessment and Accreditation of Laboratory Animal Care-approved facilities and NIH Guide for the Care and Use of Laboratory Animals.

As previously described [69], maternal morphometrics were determined pre-pregnancy to guarantee weight consistency and general morphometrics in the animals used in the present study. Non-pregnant outbred female baboons (*Papio* spp.) of a similar morphometric phenotype were selected for the study. Animals were housed at the Southwest National Primate Research Center at the Texas Biomedical Research Institute (TBRI) in the Association for Assessment and Accreditation of Laboratory Animal Care (AAALAC)-approved facilities. Housing conditions and animal caging were performed as previously described [12].

Each baboon’s weight was obtained while crossing an electronic scale (GSE 665; GSE Scale Systems, Milwaukee, WI, USA), and as described elsewhere [12].

#### 4.1.2. Experimental Design

Normally cycling female baboons from 8 to 15 years old were observed twice a day for well-being and three times a week for turgescence (genital organ’s skin swelling) and signs of vaginal bleeding to assess their reproductive cycle and enable determining the timing of pregnancy [69]. After a 30-day adaptation to the feeding system, a fertile male was introduced into each breeding cage. On day 30 of pregnancy, which was determined by following the changes in the swelling of the sex skin and by ultrasonography, twenty-four female baboons were randomly assigned to eat standard primate chow ad libitum (control diet) or to receive 70% of the average daily amount of food eaten by the female control baboons (MNR group) on a body weight-adjusted basis at same gestational age. Cesarean section was performed at 90 days gestation (0.5 G) (Figure 6). Each fetus from a singleton pregnant female baboon is considered an experimental unit; in some cases, the pregnant female baboon was also assumed as the experimental unit when maternal data are presented (12 baboons/dietary group; 6 male control fetuses—C-M; 6 female control fetuses—C-F; 6 MNR male fetuses—MNR-M; and 6 MNR female fetuses—MNR-F).

Purina Monkey Diet 5038, standard biscuits were provided once a day. The biscuit is described as a “complete life-cycle diet for all Old-World Primates” and contains stabilized vitamin C as well as all other required vitamins. The basic composition includes crude protein (≥15%), crude fat (≥5%), crude fiber (≤6%), ash (≤5%), and added minerals (≤3%) [69].

After the confirmation of pregnancy, food intake was recorded in 8 female baboons fed ad libitum and was calculated as 50.61 ± 3.61 kcal/kg of body weight per day. Before the start of the controlled diet, baboons were fed the same diet without a biscuit limit. Water was continuously available in the feeding cages via individual waterers (Lixit, Napa, CA, USA) and at several locations in the group housing. Animal food consumption, weights, and health status were recorded daily. More details regarding housing and environmental enrichment have been previously published [69].

#### 4.1.3. Cesarean Section, Fetal and Maternal Morphometry, and Blood Sampling

Mothers were fasted from their last feeding time the day before, until the cesarean section [69]. A fully certified M.D. or D.V.M. performed surgical procedures, and postsurgical care was prescribed and monitored by an accredited veterinarian. Cesarean section and fetal necropsy were performed under isoflurane anesthesia (2%, 2 L/min oxygen, tracheal intubation), followed by tranquilization with ketamine hydrochloride (10 mg/kg intramuscularly injection) at 90 days of gestation (0.5 G) using standard sterile techniques as previously described [69]. Following hysterotomy, fetal exsanguination was performed with maternal and fetal baboons under general anesthesia as approved by the American Veterinary Medical Association Panel on Euthanasia [69]. Fetal hearts were collected. Cardiac samples were taken from the free wall of the left cardiac ventricle that was cut transversely. Some pieces were flash-frozen and stored at −80 °C until analysis. Postoperatively, mothers were placed in individual cages and observed until they were upright under their power and returned to their group cage. More details regarding maternal post-partum handling have been previously described [12].

### 4.2. Analysis of mtDNA Copy Number via Quantitative Real-Time PCR

DNA extraction was performed as previously described [12]. RT-PCR was performed using the SsoFast Eva Green Supermix (Bio-Rad, Hercules, CA, USA), in a CFX96 real-time PCR system (Bio-Rad), with the primers for *ND1* (accession code NC_001992.1; sense sequence CCTATGAATCCGAGCAGCGT; antisense sequence GCTGGAGATTGCGATGGGTA) and for *B2M* (accession code NC_018158.1, sense sequence CAGGGCCCAGGACAGTTAAG; antisense sequence GGGATGGGACTCATTCAGGG) at 500 nM each. The amplification of 25 ng DNA was performed with an initial cycle of 2 min at 98 °C, followed by 40 cycles of 5 s at 98 °C plus 5 s at 60 °C. At the end of each cycle, Eva Green fluorescence was recorded to allow Ct determination. For quality control, the melting temperature of the PCR products was determined after amplification by performing melting curves, and no template controls were run.

For absolute quantification and amplification efficiency, standards at known copy numbers were produced by purifying PCR products. After optimizing the annealing temperature, products were amplified for each primer pair using the HotstarTaq Master Mix Kit (#203445 Qiagen, Hilden, Germany). Briefly, 1 μL of a DNA sample was added to a PCR tube containing the HotStar Taq Master Mix and the specific primers and placed in a CFX96 real-time PCR system. The amplification protocol started with an initial activation step of 15 min at 95 °C degrees, followed by 35 cycles of 1 min at 94 °C (denaturation) plus 1 min at 60 °C (annealing), plus 1 min at 72 °C (extension), and a final extension step of 10 min at 72 °C. After amplification, the products were purified using the MiniElute PCR purification kit (#280006 Qiagen) following the manufacturer’s instructions. Eluted DNA was quantified in a Nanodrop 2000 device, the copy numbers were adjusted to 5 × 10^9^ copies/μL, and tenfold serial dilutions were prepared. mtDNA copy number was determined by the ratio between the absolute amounts of mitochondrial gene *ND1* versus the absolute amount of the *B2M* nuclear gene in each sample, using the CFX96 Manager software (v. 3.0; Bio-Rad).

### 4.3. Gene Expression Analysis by PCR Array

RNA extraction was performed following the protocol previously described by Cox et al. [70]. RNA was quantified spectrophotometrically using Thermo Scientific NanoDrop 2000 spectrophotometer (ThermoFisher Scientific, Waltham, MA, USA) and stored at −80 °C. The RNA purity and quality were checked by Ultraviolet spectrophotometry as described elsewhere [12]. After RNA preparation, the samples were treated as previously described [12]. The RT^2^ Profiler polymerase chain reaction (PCR) Array System (SuperArray Bioscience (Frederick, MD, USA), SA Biosciences (Frederick, MD, USA), Qiagen (Hilden, Germany)), was used to evaluate the different cardiac mitochondrial transcripts between control and MNR fetuses as previously described [12]. Each PCR array contained 84 transcripts of the corresponding signaling pathway, a set of five reference genes as internal controls, and additional controls for efficiency of reverse transcription, PCR, and the absence of contaminating genomic DNA. Data were normalized with three endogenous controls that did not differ between groups hypoxanthine phosphoribosyltransferase 1 (*HPRT1*), ribosomal protein L13a (*RPL13A*), and Beta-actin (*ACTB*) and analyzed with the ΔΔCt method (where Ct is threshold cycle) using the PCR Array Data Analysis Web Portal (SA Biosciences). The transcripts used in this study are listed in Table 4 and Table 5.

### 4.4. Protein Analyses via Western Blotting

Protein analyses via Western blotting were performed following standard protocols [71]. Sample preparation and protein extraction were performed according to previously described protocols [12]. Extracted proteins were solubilized to achieve a concentration of 1 mg/mL or 2 mg/mL of protein with Laemmli buffer (62.5 mM Tris pH 6.8 (HCl), 50% glycerol, 2% SDS, 0.005% bromophenol blue, supplemented with 5% β-mercaptoethanol) and boiled for 5 min in a water bath and then centrifuged at 14,000× *g* for 5 min. Equivalent amounts of total protein (10 μg per lane) were loaded in a 10–20% gradient Tris-HCl polyacrylamide gel as well as two different standards for molecular weight estimation and for monitoring electrophoresis progress, the Precision Plus Protein Dual Color Standards (Bio-Rad) and the SeeBlue Plus2 Pre-Stained Standard (ThermoFisher Scientific (Waltham, MA, USA), Invitrogen (Waltham, MA, USA)). Electrophoresis was carried out at room temperature in a Criterion system (Bio-Rad) using 150 V until the sample buffer (blue) reaches the bottom of the gel (≈90 min). After separation by SDS-PAGE, proteins were electrophoretically transferred in a TransBlot Cell system (Bio-Rad) to a polyvinylidene difluoride (PVDF) membrane previously activated, a constant amperage (0.5 A) for 2 h at 4 °C using a CAPS transfer buffer (10 mM 3-(Cyclohexylamino)-1-propanesulfonic acid pH 11 (NaOH), 10% methanol). The quality of the electrophoretic transfer was evaluated by the complete transfer of pre-stained molecular weight markers below 100 kDa and via Ponceau staining. Ponceau results were also used to confirm an equal amount of protein loading and to normalize band density. After Ponceau removal, the membranes were blocked in 5% non-fat milk/PBS overnight at 4 °C with agitation. Before incubation with primary antibodies, the membrane was washed for 10 min in PBS 0.05% Tween-20 (PBS-T). Primary antibodies were prepared in 1% non-fat milk/PBS to a final volume of 5 mL and incubated overnight at 4 °C. After incubation with primary antibodies, membranes were washed with PBS-T solution three times, 5 min each, and incubated with the correspondent alkaline phosphatase-conjugated secondary antibodies for 2 h at room temperature with stirring. For immunodetection, membranes were washed three times for 5 min each with PBS-T, rinsed in PBS to remove any Tween-20, which can be inhibitory to the detection method, dried, and incubated with an enhanced chemifluorescence (ECF) system (#RPN5785, GE Healthcare, Little Chalfont, Buckinghamshire, UK) during a maximum of 5 min. Density analysis of bands was carried out with VisionWorks LS Image Acquisition and Analysis Software (UVP). The resulting images were analyzed and densities were normalized to Ponceau. The average value of the C-Males (M) group was assumed as one unit, and the values of each sample were determined proportionally. All of the primary antibodies used in this experiment were purchased from abcam: NDUFB8 (1:500, ab110242), UQCRC2 (1:500, ab14745), MT-CO2 (1:500, ab110258), COX6C (1:1000, ab150422), ATP5A (1:500, ab110273), TOMM20 (1:500, sc11415), TFAM (1:00, sc23588), and citrate synthase (1:1000, ab129088). All of the secondary antibodies were purchased from Santa Cruz and were used in a 1:5000 dilution: rabbit (sc-2007), mouse (sc-2008), goat (sc-2771).

### 4.5. Data Analysis and Statistics

The software GraphPad Prism version 8.0 (GraphPad Software, San Diego, CA, USA) was used for data analysis. Each pregnant baboon and the corresponding fetus were considered an experimental unit. Outbred pregnant female baboons were randomly assigned to control or MNR groups. Data are expressed as mean or as mean ± SD. Normality was assessed via the Kolmogorov–Smirnov or Shapiro–Wilk tests. To assess the effect of MNR on fetal cardiac mitochondrial parameters, the following comparisons were made: (1) only to evaluate diet-induced alterations, independently of fetal sex: between Control (C) vs. MNR; (2) to simultaneously assess the impact of diet and sex: between male control (C-M) vs. female control (C-F), male MNR (MNR-M) vs. C-M, female MNR (MNR-F) vs. C-F, and MNR-M vs. MNR-F. The orange software (version 3.32.0) was used for the computational data analysis and visualization. Clustering (opt ordering) was applied to both columns and rows. Data were normalized for value ranges between −1 and 1.

## 5. Conclusions

In this study, 30% MNR led to sex-specific alterations on the NHP fetal cardiac left-ventricle mitochondrial-related gene expression at 90 days old (0.5 G), some of which persisted until near term (0.9 G). More than half of the MNR-induced gene expression alterations encode for OXPHOS complex subunits. Given that the OXPHOS system is primarily responsible for energy production in the mitochondria, early IUGR-induced mitochondrial adaptations could play a role in IUGR offspring’s increased predisposition to CVD. The fact that we here showed that MNR can induce fetal cardiac mitochondrial function alterations as early as the mid-gestation period highlights the need to better understand the early origins of CVD, which can provide new targets for disease prevention and mitigation.

## Figures and Tables

**Figure 1 ijms-24-15192-f001:**
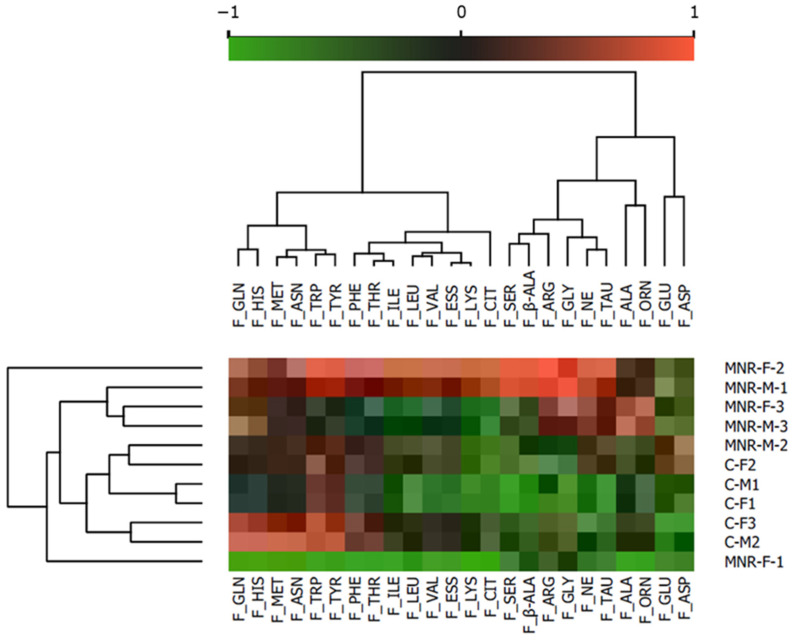
Plasma amino acid profile in 90-day-old control (C) fetuses and fetuses born from maternal nutrient reduction (MNR) conditions. Heatmap representation of plasmatic amino acid variation: in Control (C): fetuses born from mothers fed a control diet; in MNR: fetuses born from mothers fed a 30% nutrient reduction diet. Orange software (version 3.32.0) was used for the computational data analysis and visualization. Clustering (opt ordering) was applied to both columns and rows. Data were normalized for value ranges between −1 and 1. F_GLN—Glutamine; F_HIS—Histidine; F_MET—Methionine; F_ASN—Asparagine; F_TRP—Tryptophane; F_TYR—Tyrosine; F_PHE—Phenylalanine; F_THR—Threonine; F_ILE—Isoleucine; F_LEU—Leucine; F_VAL—Valine; F_ESS—Total essential; F_LYS—Lysine; F_CIT—Citrulline; F_SER—Serine; F_β-ALA—β-Alanine; F_ARG—Arginine; F_GLY—Glycine; F_NE—Non-essential; F_TAU—Taurine; F_ALA—Alanine; F_ORN—Ornithine; F_GLU—Glutamate; F_ASP—Aspartame.

**Figure 2 ijms-24-15192-f002:**
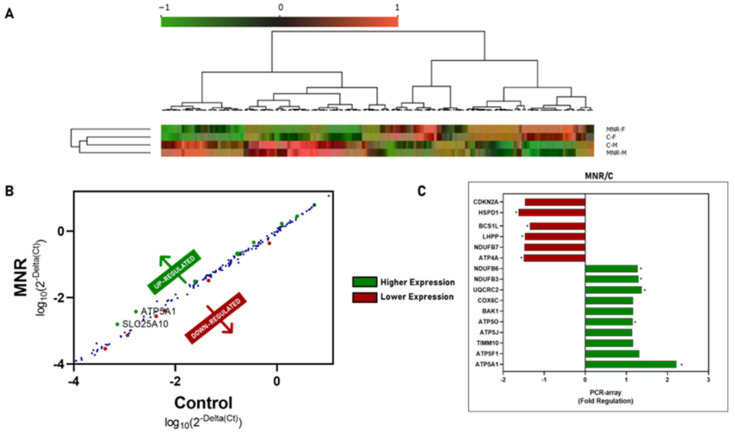
Cardiac left ventricle (LV) tissue gene expression of 90-day-old control (C) fetuses and fetuses born from maternal nutrient reduction (MNR) conditions. (**A**) Gene expression heatmap view of Control fetuses born from mothers fed a control diet and MNR fetuses born from mothers fed a 30% nutrient-reduced diet; (**B**) differences in gene expression of fetal cardiac left ventricle (LV) tissue of fetuses in MNR conditions relative to the Control group, dots represent the plot between log_10_(2^−Delta(Ct)^) for the control group and MNR for each gene represented in Table 3, in blue—unaltered gene expression, in red—lower gene expression, in green—higher gene expression; (**C**) variation in fold-regulation of gene expression in fetal cardiac left ventricle tissue of fetuses in MNR conditions relative to the Control group. PCR arrays were used to evaluate mRNA abundance of mitochondrial transcripts. Values were normalized to endogenous controls (hypoxanthine phosphoribosyltransferase 1 (*HPRT1*), ribosomal protein L13a (*RPL13A*), and Beta-actin (*ACTB*)) and are expressed relative to their normalized values. The mean of gene expression for each group is represented (*n* ≥ 3 per group). Orange software package (version 3.32.0) was used for the computational data analysis and visualization. Clustering (opt ordering) was applied to both columns and rows. Data were normalized for value ranges between −1 and 1. The clustering and heatmap analysis highlight the transcriptional changes. The fold-regulation was calculated between the normalized gene expression of MNR samples and the normalized gene expression of control samples. All of the shown transcripts have a *p*-value < 0.1 vs. the control group. Comparison between groups was evaluated using a non-parametric Mann–Whitney test. The mean of gene expression is represented (*n* ≥ 3 per group). A *p*-value ≤ 0.05 was considered statistically significant (*), and gene expression was considered altered (upregulated or downregulated when comparing C vs. MNR).

**Figure 3 ijms-24-15192-f003:**
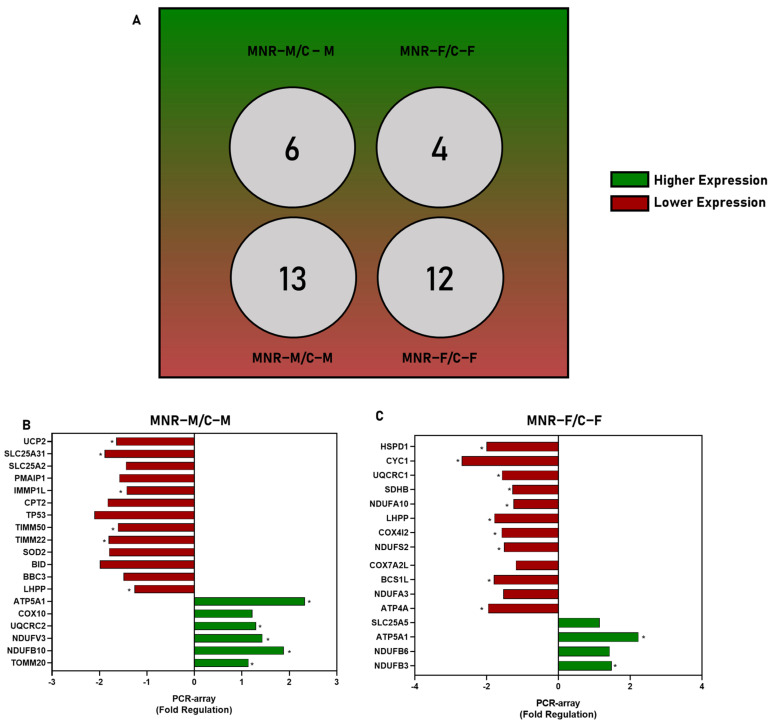
Fold-regulation of 90-day-old fetal cardiac left ventricle (LV) tissues in fetuses in maternal nutrient reduction (MNR) conditions relative to the Control (C) group of the respective sex. Control (C): fetuses born from mothers fed a control diet; MNR: fetuses born from mothers fed a 30% nutrient-restricted diet; C-M/C-F: male/female fetuses born from mothers fed a control diet; MNR-M/F: fetuses born from mothers fed a 30% nutrient-restricted diet. (**A**) Diagram representing the number of genes from LV of MNR fetuses that were upregulated and downregulated relative to the C group of the same sex; (**B**,**C**) Fold-regulation of gene expression analysis of fetal cardiac LV tissue from MNR conditions relative to the C group of the respective sex. PCR arrays were used to evaluate mRNA abundance of mitochondrial transcripts. Values were normalized to endogenous controls (hypoxanthine phosphoribosyltransferase 1 (*HPRT1*), ribosomal protein L13a (*RPL13A*), and Beta-actin (*ACTB*)) and are expressed relative to their normalized values. The fold-regulation was calculated between the normalized gene expression of MNR samples and the normalized gene expression of control samples, according to sex ((**B**) MNR-M vs. C-M; (**C**) MNR-F vs. C-F)). All of the present transcripts have a *p*-value < 0.1 vs. the control group. Comparison between groups was evaluated using a non-parametric Mann–Whitney test (*n* ≥ 3 per group)). The mean of gene expression is represented. A *p*-value ≤ 0.05 was considered statistically significant (*), and gene expression was considered altered (upregulated or downregulated).

**Figure 4 ijms-24-15192-f004:**
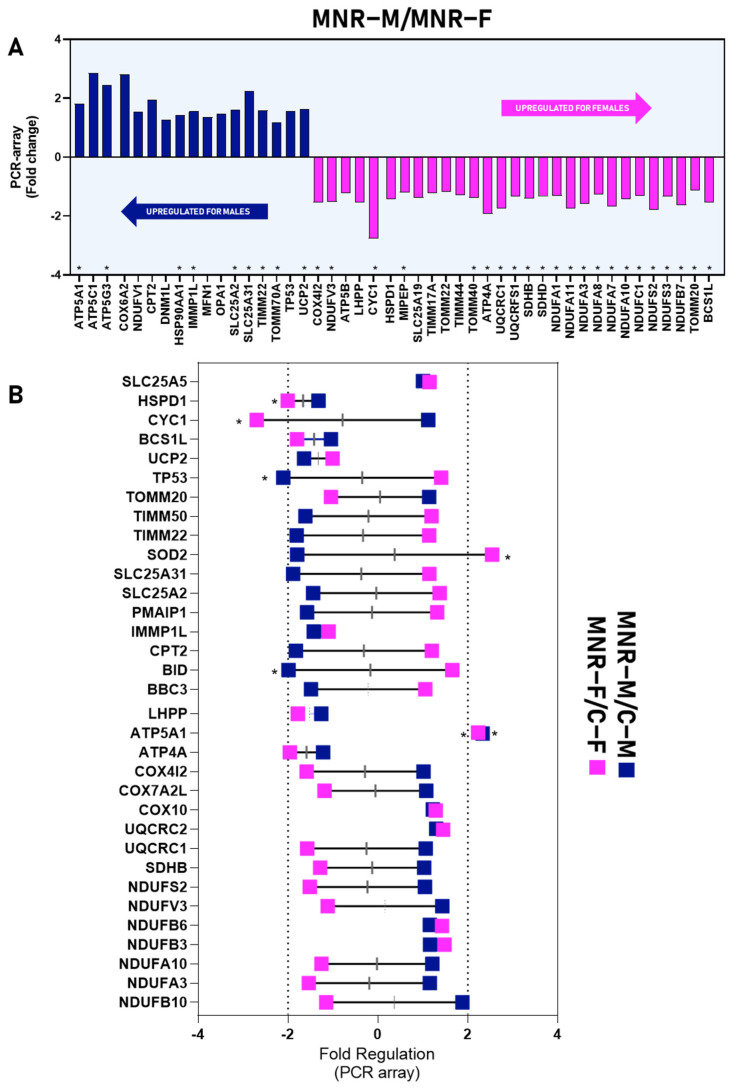
Sex-specific differences in gene expression fold-regulation of 90-day-old fetal cardiac left ventricle (LV) tissue. Control (C): fetuses born from mothers fed a control diet; MNR: fetuses born from mothers fed a 30% nutrient-restricted diet; C-M/C-F: male/female fetuses born from mothers fed a control diet; MNR-M/F: fetuses born from mothers fed a 30% nutrient-restricted diet. Fold-regulation of gene expression analysis of fetal cardiac LV tissue from control and MNR groups, according to fetal sex. PCR arrays were used to evaluate mRNA abundance of mitochondrial transcripts. Values were normalized to endogenous controls (hypoxanthine phosphoribosyltransferase 1 (*HPRT1*), ribosomal protein L13a (*RPL13A*), and Beta-actin (*ACTB*)) and are expressed relative to their normalized values. (**A**) The fold-regulation was calculated between the normalized gene expression of MNR-M samples and the normalized gene expression of MNR-F samples; (**B**) The fold-regulation was calculated between the normalized gene expression of MNR-M samples and the normalized gene expression of C-M samples (represented in blue) or between the normalized gene expression of MNR-F samples and the normalized gene expression of C-F samples (represented in pink). Comparison between groups was evaluated using a non-parametric Mann–Whitney test. The mean of gene expression is represented (*n* ≥ 3 per group). A *p*-value ≤ 0.05 was considered statistically significant (*), and gene expression was considered altered (upregulated or downregulated).

**Figure 5 ijms-24-15192-f005:**
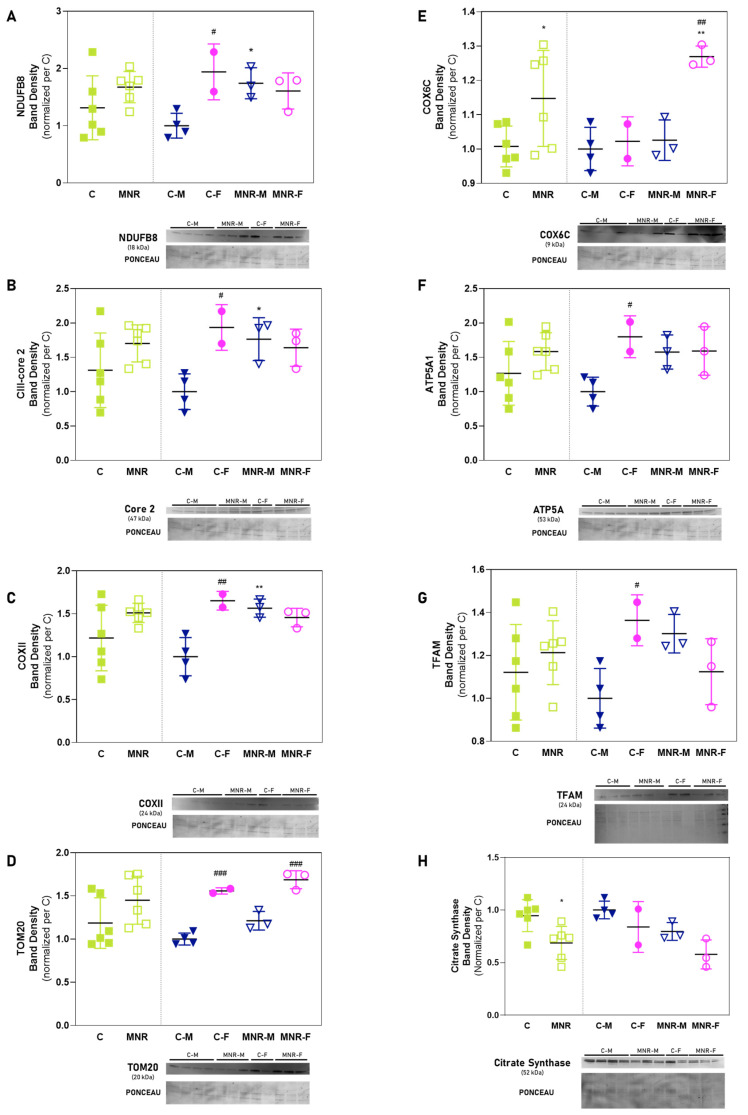
Cardiac relative band density of mitochondrial function-associated proteins in the left ventricles of 90-day-old fetuses. Control (C): fetuses born from mothers fed a control diet; MNR: fetuses born from mothers fed a 30% nutrient-restricted diet; C-M/C-F: male/female fetuses born from mothers fed a control diet (filled symbols); MNR-M/MNR-F: fetuses born from mothers fed a 30% nutrient-restricted diet (open symbols). (**A**) Complex I (NADH dehydrogenase) subunit NDUFB8; (**B**) complex III (cytochrome *c* reductase) subunit UQCRC2; (**C**) complex IV subunit COXII; (**D**) translocase of outer mitochondrial membrane 20 (TOMM20); (**E**) complex IV (cytochrome *c* oxidase) subunit 6C (COX6C); (**F**) ATP synthase subunit ATP5A; (**G**) mitochondrial transcription factor A (TFAM); (**H**) citrate synthase. Data are expressed as mean ± SD. Data were normalized with Ponceau S staining and the protein expression represented relative to the mean of the Control group (male). The comparison between groups was performed using a two-way ANOVA (*n* ≥ 2 per group). Normality was evaluated via Shapiro–Wilk test. Normality was evaluated via Shapiro-Wilk test. *,# *p* ≤ 0.05; **,## *p* ≤ 0.01; ### *p* ≤ 0.001; * vs. Control of the same sex; # vs. same experimental group of the opposite sex. Light green squares represent the experimental groups C and MNR with both sexes combined. Dark blue inverted triangles represent male data, and pink circles represent female data. Filled symbols correspond to the control group, while open symbols correspond to the MNR groups.

**Figure 6 ijms-24-15192-f006:**
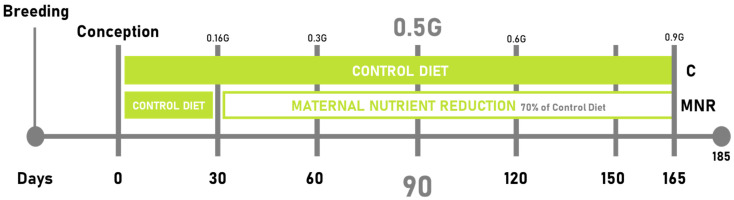
Timeline of maternal nutrition during fetal development. Control group—C; maternal nutrient reduction—MNR.

**Table 1 ijms-24-15192-t001:** Summary of gestational parameters. Maternal and fetal morphological parameters at 0.5 gestation in control pregnancies and in maternal nutrient reduction (MNR) conditions via a 30% reduction of the food eaten by control mothers on a weight-adjusted basis.

Maternal and Fetal Morphological Parameters at 0.5 Gestation																	
Maternal																	
	Combination				Mothers of Male Fetuses				Mothers of Female Fetuses				Mann–Whitney U Test				
	C		MNR		C		MNR		C		MNR		Diet	Male C vs. MNR	Female C vs. MNR	C M vs. F	MNR M vs. F
	(*n* = 11)		(*n* = 11)		(*n* = 6)		(*n* = 6)		(*n* = 5)		(*n* = 5)						
	Mean	SEM	Mean	SEM	Mean	SEM	Mean	SEM	Mean	SEM	Mean	SEM	*p*−Value	*p*-Value	*p*-Value	*p*-Value	*p*-Value
Body mass at pre-conception (kg)	13.90	0.46	14.26	0.61	13.45	0.71	14.79	1.00	14.34	0.59	13.61	0.60	–	–	–	–	–
Body mass at Cs (kg)	13.79	0.46	13.54	0.65	13.41	0.75	14.07	1.07	14.18	0.55	12.90	0.65	–	–	–	–	–
Body mass variation (kg)	−0.11	0.12	−0.72	0.15	−0.05	0.20	−0.72	0.20	−0.16	0.15	−0.71	0.24	0.008	–	0.047	–	–
Age (year)	10.79	0.94	9.86	0.82	11.27	1.83	9.84	1.45	10.30	0.72	9.88	0.74	–	–	–	–	–
**Fetal**																	
	**Combined Sex**				**Male**				**Female**				**Mann–Whitney U Test**				
	**C**		**MNR**		**C**		**MNR**		**C**		**MNR**		**Diet**	**Male** **C vs. MNR**	**Female** **C vs. MNR**	**C** **M vs. F**	**MNR** **M vs. F**
	**(*n* = 11)**		**(*n* = 11)**		**(*n* = 6)**		**(*n* = 6)**		**(*n* = 5)**		**(*n* = 5)**						
	**Mean**	**SEM**	**Mean**	**SEM**	**Mean**	**SEM**	**Mean**	**SEM**	**Mean**	**SEM**	**Mean**	**SEM**	***p*-Value**	***p*-Value**	***p*-Value**	***p*-Value**	***p*-Value**
Body mass (g)	101.14	3.25	101.19	2.96	103.32	5.55	103.67	4.69	98.52	2.98	98.22	3.35	–	–	–	–	–
Heart mass (g)	0.85	0.21	0.81	0.19	0.59	0.07	0.93	0.34	1.12	0.41	0.66	0.05	–	–	–	–	–
HW/BW (×1000)	8.31	2.01	7.84	1.59	5.49	0.61	8.82	2.96	11.14	3.72	6.66	0.33	–	–	–	0.047	–
Body length (cm)	17.41	0.31	17.52	0.27	17.00	0.41	17.50	0.29	17.90	0.43	17.54	0.52	–	–	–	–	–
BMI (kg/m^2^)	3.35	0.12	3.31	0.13	3.58	0.15	3.40	0.22	3.09	0.13	3.20	0.12	–	–	–	0.045	–
Chest circ (cm)	58.77	11.92	52.32	12.39	35.25	16.64	48.75	17.77	87.00	1.22	56.60	19.04	–	–	–	–	–
Waist circ (cm)	49.77	10.02	44.95	10.79	30.42	14.10	42.50	15.35	73.00	2.55	47.90	16.82	–	–	–	–	–
Hip circ (cm)	64.31	8.31	68.33	1.67	50.67	21.88	70.00	2.89	72.50	2.24	66.67	1.67	–	–	–	–	–
Femur (cm)	3.47	0.09	3.20	0.11	3.28	0.10	3.33	0.17	3.70	0.09	3.04	0.13	–	–	0.014	0.021	–

**Table 2 ijms-24-15192-t002:** Maternal and fetal blood biochemical parameters at 50% gestation in control pregnancies and in the presence of 30% maternal nutrient restriction (MNR) of the food eaten by control mothers on a weight-adjusted basis.

Comparison of Maternal and Fetal Blood Biochemical Parameters at 0.5 G																	
Maternal																	
	Combination				Mothers of Male Fetuses				Mothers of Female Fetuses				Mann–Whitney U Test				
	C		MNR		C		MNR		C		MNR		Diet	Male C vs. MNR	Female C vs. MNR	C M vs. F	MNR M vs. F
	(*n* = 8)		(*n* = 6)		(*n* = 3)		(*n* = 3)		(*n* = 5)		(*n* = 3)						
	Mean	SEM	Mean	SEM	Mean	SEM	Mean	SEM	Mean	SEM	Mean	SEM	*p*-Value	*p*-Value	*p*-Value	*p*-Value	*p*-Value
Hemoglobin (g/dL)	12.61	0.34	12.90	0.44	12.73	0.46	13.27	0.15	12.54	0.51	12.53	0.90	–	–	–	–	–
Glucose	74.38	9.63	85.83	10.63	55.67	2.91	97.67	13.22	85.60	13.15	74.00	15.82	–	0.05		0.04	
BUN	10.25	0.67	7.50	0.43	8.67	0.67	7.33	0.67	11.20	0.73	7.67	0.67	0.01	–	0.02	0.04	
BUN/CREAT Ratio	11.59	1.40	8.37	0.61	12.30	3.86	7.13	0.43	11.16	0.95	9.60	0.40	–	–	–	–	0.04
Cholesterol	51.63	5.64	42.17	1.70	44.33	9.26	40.00	2.00	56.00	7.11	44.33	2.40	–	–	–	–	–
Total Protein	6.70	0.19	6.68	0.16	6.73	0.30	6.73	0.33	6.68	0.27	6.63	0.15	–	–	–	–	–
Sodium	140.00	0.98	142.83	1.01	138.67	1.86	143.67	1.76	140.80	1.11	142.00	1.15	0.05	–	–	–	–
Potassium	3.59	0.10	3.92	0.21	3.37	0.19	4.03	0.18	3.72	0.09	3.80	0.42	–	0.05	–	–	–
Triglycerides	28.88	3.36	30.00	6.12	26.67	4.63	33.67	11.67	30.20	4.90	26.33	6.12	–	–	–	–	–
**Fetal**																	
	**Combined Sex**				**Male**				**Female**				**Mann–Whitney U Test**				
	**C**		**MNR**		**C**		**MNR**		**C**		**MNR**		**Diet**	**Male** **C vs. MNR**	**Female** **C vs. MNR**	**C** **M vs. F**	**MNR** **M vs. F**
	**(*n* = 8)**		**(*n* = 6)**		**(*n* = 3)**		**(*n* = 3)**		**(*n* = 5)**		**(*n* = 3)**						
	**Mean**	**SEM**	**Mean**	**SEM**	**Mean**	**SEM**	**Mean**	**SEM**	**Mean**	**SEM**	**Mean**	**SEM**	***p*-Value**	***P*-Value**	***p*-Value**	***p*-Value**	***p*-Value**
Hemoglobin (g/dL)	11.08	0.21	11.77	0.15	11.37	0.09	.	.	10.65	0.35	11.77	0.15	0.04	–	–	–	–
Glucose	47.14	4.35	39.00	7.44	34.50	7.50	43.33	15.84	52.20	3.43	34.67	2.73	–	–	0.04	–	–
BUN	11.00	0.58	8.00	0.52	9.50	0.50	7.33	0.67	11.60	0.60	8.67	0.67	0.00	–	0.02	0.03	–
BUN/CREAT Ratio	13.97	0.96	9.97	0.98	13.35	3.35	8.50	0.76	14.22	0.88	11.43	1.43	0.04	–	–	–	–
Cholesterol	65.57	4.06	58.50	3.43	65.00	3.00	58.00	4.93	65.80	5.80	59.00	5.86	–	–	–	–	–
Total Protein	2.60	0.06	2.65	0.04	2.60	0.00	2.67	0.03	2.60	0.09	2.63	0.09	–	–	–	–	–
Sodium	137.71	1.66	141.40	0.40	134.50	6.50	141.00	0.00	139.00	0.32	141.67	0.67	0.01	–	0.02	–	–
Potassium	4.03	0.28	4.36	0.40	3.50	0.20	4.20	0.90	4.24	0.35	4.47	0.50	–	–	–	–	–
Triglycerides	36.43	4.94	28.83	5.53	37.00	8.00	25.00	4.04	36.20	6.69	32.67	11.05	–	–	–	–	–

**Table 3 ijms-24-15192-t003:** Fold-regulation of gene expression analysis of 90-day-old fetal cardiac left ventricle (LV) tissue. Control (C): fetuses born from mothers fed a control diet; MNR: fetuses born from mothers fed a 30% nutrient-reduced diet; C-M/C-F: male/female fetuses born from mothers fed a control diet; MNR-M/F: fetuses born from mothers fed a 30% nutrient-reduced diet. PCR arrays were used to evaluate mRNA abundance of mitochondrial transcripts. Values were normalized to endogenous controls (hypoxanthine phosphoribosyltransferase 1 (*HPRT1*), ribosomal protein L13a (*RPL13A*), and Beta-actin (*ACTB*)) and are expressed relative to their normalized values. In the first group, the fold-regulation was calculated between the normalized gene expression of MNR samples and the normalized gene expression of control samples. In the second and third group of analysis, the fold-regulation was calculated between the normalized gene expression of MNR samples and the normalized gene expression of control samples of the respective sex (MNR-M vs. C-M and MNR-F vs. C-F). In the fourth group, the fold-regulation was calculated between the normalized gene expression of MNR-M samples and the normalized gene expression of MNR-F samples. Comparison between groups was evaluated using a non-parametric Mann–Whitney test (*n* ≥ 2). The mean of gene expression is represented. A *p*-value ≤ 0.05 was considered statistically significant, and gene expression was considered altered (upregulated or downregulated).

Gene Expression Fold Regulation and *p*-Value			
Group	Gene	Fold Regulation	*p*-Value
MNR vs. C	*CDKN2A*	–	0.0558
	*HSPD1*	−1.6229	0.0097
	*BCS1L*	−1.3446	0.0089
	*LHPP*	−1.4705	0.0026
	*NDUFB7*	–	0.0587
	*ATP4A*	−1.5006	0.0199
	*NDUFB6*	1.2714	0.0474
	*NDUFB3*	1.2952	0.0125
	*UQCRC2*	1.3692	0.0274
	*COX6C*	–	0.0714
	*BAK1*	–	0.0856
	*ATP5O*	1.151	0.0373
	*ATP5J*	–	0.0588
	*TIMM10*	–	0.0762
	*ATP5F1*	–	0.0595
	*ATP5A1*	2.2	0.0089
MNR-M vs. C-M	*UCP2*	−1.6445	0.0155
	*SLC25A31*	−1.8892	0.0416
	*SLC25A2*	–	0.0677
	*PMAIP1*	–	0.0770
	*IMMP1L*	−1.4269	0.0070
	*CPT2*	–	0.0958
	*TP53*	–	0.0577
	*TIMM50*	−1.6130	0.0319
	*TIMM22*	−1.8063	0.0223
	*SOD2*	–	0.0620
	*BID*	–	0.0759
	*BBC3*	–	0.0848
	*LHPP*	−1.2608	0.0369
	*ATP5A1*	2.3	0.0004
	*COX10*	–	0.0827
	*UQCRC2*	1.2991	0.0298
	*NDUFV3*	1.4313	0.0382
	*NDUFB10*	1.8811	0.0035
	*TOMM20*	1.1394	0.0481
MNR-F vs. C-F	*HSPD1*	−2.0	0.0108
	*CYC1*	−2.7	0.0039
	*UQCRC1*	1.5758	0.0089
	*SDHB*	−1.2856	0.0066
	*NDUFA10*	−1.2561	0.0236
	*LHPP*	−1.7836	0.0142
	*COX4I2*	−1.5859	0.0211
	*NDUFS2*	−1.5158	0.0022
	*COX7A2L*	–	0.0671
	*BCS1L*	−1.7991	0.0015
	*NDUFA3*	–	0.0942
	*ATP4A*	−1.9599	0.0182
	*SLC25A5*	–	0.0868
	*ATP5A1*	2.2	0.0419
	*NDUFB6*	–	0.0974
	*NDUFB3*	1.4818	0.0207
MNR-M vs. MNR-F	*ATP5A1*	1.8097	0.0446
	*ATP5C1*	–	0.0996
	*ATP5G3*	2.4323	0.0173
	*COX6A2*	–	0.0668
	*NDUFV1*	–	0.0624
	*CPT2*	–	0.0539
	*DNM1L*	–	0.0999
	*HSPD90AA1*	1.4362	0.0029
	*IMMP1L*	1.5493	0.0042
	*MFN1*	–	0.0924
	*OPA1*	–	0.0655
	*SLC25A2*	1.5927	0.0175
	*SLC25A31*	2.2323	0.0154
	*TIMM22*	1.5782	0.0229
	*TOMM70A*	1.1683	0.0102
	*TP53*	–	0.0850
	*UCP2*	1.6187	0.0465
	*COX4I2*	−1.5388	0.0368
	*NDUFV3*	−1.5110	0.0201
	*ATP5B*	–	0.0588
	*LHPP*	−1.5396	0.0255
	*CYC1*	−2.7650	0.0036
	*HSPD1*	–	0.0591
	*MIPEP*	−1.1997	0.0007
	*SLC25A19*	–	0.0997
	*TIMM17A*	–	0.0961
	*TOMM22*	–	0.0823
	*TIMM44*	–	0.0885
	*TOMM40*	−1.3842	0.0427
	*ATP4A*	−1.9241	0.0034
	*UQCRC1*	−1.7358	0.0011
	*UQCRFS1*	−1.3300	0.0213
	*SDHB*	−1.4021	0.0053
	*SDHD*	−1.3393	0.0046
	*NDUFA1*	−1.3118	0.0246
	*NDUFA11*	−1.7337	0.0439
	*NDUFA3*	−1.5965	0.0239
	*NDUFA8*	−1.2592	0.0271
	*NDUFA7*	−1.6862	0.004
	*NDUFA10*	−1.4186	0.0113
	*NDUFC1*	−1.3032	0.0121
	*NDUFS2*	−1.7828	0.0039
	*NDUFS3*	−1.3326	0.0118
	*NDUFB7*	−1.6299	0.0388
	*TOMM20*	−1.1220	0.0320
	*BCS1L*	−1.5516	0.0052

**Table 4 ijms-24-15192-t004:** Panel of gene expression analyzed using the Human Mitochondrial Energy Metabolism RT^2^ Profiler PCR Array. This array profiled the expression of 84 key genes involved in mitochondrial energy metabolism, including genes encoding components of the electron transport chain and oxidative phosphorylation complexes. Position indicates the location in the 96-well plate where the transcripts were assessed, Symbol denotes the gene identification, RefSeq denotes the Reference Sequence from the National Center for Biotechnology Information collection, and Description gives summary information about the gene identification and/or function.

Position	Symbol	Refseq	Description
A01	*ATP12A*	NM_001676	ATPase, H^+^/K^+^ transporting, nongastric, alpha polypeptide
A02	*ATP4A*	NM_000704	ATPase, H^+^/K^+^ exchanging, alpha polypeptide
A03	*ATP4B*	NM_000705	ATPase, H^+^/K^+^ exchanging, beta polypeptide
A04	*ATP5A1*	NM_004046	ATP synthase, H^+^ transporting, mitochondrial F1 complex, alpha subunit 1
A05	*ATP5B*	NM_001686	ATP synthase, H^+^ transporting, mitochondrial F1 complex, beta polypeptide
A06	*ATP5C1*	NM_005174	ATP synthase, H^+^ transporting, mitochondrial F1 complex, gamma polypeptide 1
A07	*ATP5F1*	NM_001688	ATP synthase, H^+^ transporting, mitochondrial Fo complex, subunit B1
A08	*ATP5G1*	NM_005175	ATP synthase, H^+^ transporting, mitochondrial Fo complex, subunit C1
A09	*ATP5G2*	NM_001002031	ATP synthase, H^+^ transporting, mitochondrial Fo complex, subunit C2
A10	*ATP5G3*	NM_001689	ATP synthase, H^+^ transporting, mitochondrial Fo complex, subunit C3
A11	*ATP5H*	NM_006356	ATP synthase, H^+^ transporting, mitochondrial Fo complex, subunit d
A12	*ATP5I*	NM_007100	ATP synthase, H^+^ transporting, mitochondrial Fo complex, subunit E
B01	*ATP5J*	NM_001685	ATP synthase, H^+^ transporting, mitochondrial Fo complex, subunit F6
B02	*ATP5J2*	NM_004889	ATP synthase, H^+^ transporting, mitochondrial Fo complex, subunit F2
B03	*ATP5L*	NM_006476	ATP synthase, H^+^ transporting, mitochondrial Fo complex, subunit G
B04	*ATP5O*	NM_001697	ATP synthase, H^+^ transporting, mitochondrial F1 complex, O subunit
B05	*ATP6V0A2*	NM_012463	ATPase, H^+^ transporting, lysosomal V0 subunit a2
B06	*ATP6V0D2*	NM_152565	ATPase, H^+^ transporting, lysosomal 38 kDa, V0 subunit d2
B07	*ATP6V1C2*	NM_144583	ATPase, H^+^ transporting, lysosomal 42 kDa, V1 subunit C2
B08	*ATP6V1E2*	NM_080653	ATPase, H^+^ transporting, lysosomal 31 kDa, V1 subunit E2
B09	*ATP6V1G3*	NM_133262	ATPase, H^+^ transporting, lysosomal 13 kDa, V1 subunit G3
B10	*BCS1L*	NM_004328	BCS1-like (S. cerevisiae)
B11	*COX4I1*	NM_001861	Cytochrome c oxidase subunit IV isoform 1
B12	*COX4I2*	NM_032609	Cytochrome c oxidase subunit IV isoform 2
C01	*COX5A*	NM_004255	Cytochrome c oxidase subunit Va
C02	*COX5B*	NM_001862	Cytochrome c oxidase subunit Vb
C03	*COX6A1*	NM_004373	Cytochrome c oxidase subunit VIa polypeptide 1
C04	*COX6A2*	NM_005205	Cytochrome c oxidase subunit VIa polypeptide 2
C05	*COX6B1*	NM_001863	Cytochrome c oxidase subunit Vib polypeptide 1
C06	*COX6B2*	NM_144613	Cytochrome c oxidase subunit VIb polypeptide 2
C07	*COX6C*	NM_004374	Cytochrome c oxidase subunit Vic
C08	*COX7A2*	NM_001865	Cytochrome c oxidase subunit VIIa polypeptide 2
C09	*COX7A2L*	NM_004718	Cytochrome c oxidase subunit VIIa polypeptide 2 like
C10	*COX7B*	NM_001866	Cytochrome c oxidase subunit VIIb
C11	*COX8A*	NM_004074	Cytochrome c oxidase subunit VIIIA
C12	*COX8C*	NM_182971	Cytochrome c oxidase subunit VIIIC
D01	*CYC1*	NM_001916	Cytochrome c-1
D02	*LHPP*	NM_022126	Phospholysine phosphohistidine inorganic pyrophosphate phosphatase
D03	*NDUFA1*	NM_004541	NADH dehydrogenase (ubiquinone) 1 alpha subcomplex, 1
D04	*NDUFA10*	NM_004544	NADH dehydrogenase (ubiquinone) 1 alpha subcomplex, 10
D05	*NDUFA11*	NM_175614	NADH dehydrogenase (ubiquinone) 1 alpha subcomplex, 11
D06	*NDUFA2*	NM_002488	NADH dehydrogenase (ubiquinone) 1 alpha subcomplex, 2
D07	*NDUFA3*	NM_004542	NADH dehydrogenase (ubiquinone) 1 alpha subcomplex, 3
D08	*NDUFA4*	NM_002489	NADH dehydrogenase (ubiquinone) 1 alpha subcomplex, 4
D09	*NDUFA5*	NM_005000	NADH dehydrogenase (ubiquinone) 1 alpha subcomplex, 5
D10	*NDUFA6*	NM_002490	NADH dehydrogenase (ubiquinone) 1 alpha subcomplex, 6
D11	*NDUFA7*	NM_005001	NADH dehydrogenase (ubiquinone) 1 alpha subcomplex, 7
D12	*NDUFA8*	NM_014222	NADH dehydrogenase (ubiquinone) 1 alpha subcomplex, 8
E01	*NDUFAB1*	NM_005003	NADH dehydrogenase (ubiquinone) 1, alpha/beta subcomplex, 1
E02	*NDUFB10*	NM_004548	NADH dehydrogenase (ubiquinone) 1 beta subcomplex, 10
E03	*NDUFB2*	NM_004546	NADH dehydrogenase (ubiquinone) 1 beta subcomplex, 2
E04	*NDUFB3*	NM_002491	NADH dehydrogenase (ubiquinone) 1 beta subcomplex, 3
E05	*NDUFB4*	NM_004547	NADH dehydrogenase (ubiquinone) 1 beta subcomplex, 4
E06	*NDUFB5*	NM_002492	NADH dehydrogenase (ubiquinone) 1 beta subcomplex, 5
E07	*NDUFB6*	NM_182739	NADH dehydrogenase (ubiquinone) 1 beta subcomplex, 6
E08	*NDUFB7*	NM_004146	NADH dehydrogenase (ubiquinone) 1 beta subcomplex, 7
E09	*NDUFB8*	NM_005004	NADH dehydrogenase (ubiquinone) 1 beta subcomplex, 8
E10	*NDUFB9*	NM_005005	NADH dehydrogenase (ubiquinone) 1 beta subcomplex, 9
E11	*NDUFC1*	NM_002494	NADH dehydrogenase (ubiquinone) 1, subcomplex unknown, 1
E12	*NDUFC2*	NM_004549	NADH dehydrogenase (ubiquinone) 1, subcomplex unknown, 2
F01	*NDUFS1*	NM_005006	NADH dehydrogenase (ubiquinone) Fe-S protein 1
F02	*NDUFS2*	NM_004550	NADH dehydrogenase (ubiquinone) Fe-S protein 2
F03	*NDUFS3*	NM_004551	NADH dehydrogenase (ubiquinone) Fe-S protein 3
F04	*NDUFS4*	NM_002495	NADH dehydrogenase (ubiquinone) Fe-S protein 4
F05	*NDUFS5*	NM_004552	NADH dehydrogenase (ubiquinone) Fe-S protein 5
F06	*NDUFS6*	NM_004553	NADH dehydrogenase (ubiquinone) Fe-S protein 6
F07	*NDUFS7*	NM_024407	NADH dehydrogenase (ubiquinone) Fe-S protein 7
F08	*NDUFS8*	NM_002496	NADH dehydrogenase (ubiquinone) Fe-S protein 8
F09	*NDUFV1*	NM_007103	NADH dehydrogenase (ubiquinone) flavoprotein 1
F10	*NDUFV2*	NM_021074	NADH dehydrogenase (ubiquinone) flavoprotein 2
F11	*NDUFV3*	NM_021075	NADH dehydrogenase (ubiquinone) flavoprotein 3
F12	*OXA1L*	NM_005015	Oxidase (cytochrome c) assembly 1-like
G01	*PPA1*	NM_021129	Pyrophosphatase (inorganic) 1
G02	*PPA2*	NM_176869	Pyrophosphatase (inorganic) 2
G03	*SDHA*	NM_004168	Succinate dehydrogenase complex, subunit A, flavoprotein (Fp)
G04	*SDHB*	NM_003000	Succinate dehydrogenase complex, subunit B, iron sulfur (Ip)
G05	*SDHC*	NM_003001	Succinate dehydrogenase complex, subunit C, integral membrane protein
G06	*SDHD*	NM_003002	Succinate dehydrogenase complex, subunit D, integral membrane protein
G07	*UQCR11*	NM_006830	Ubiquinol-cytochrome c reductase, complex III subunit XI
G08	*UQCRC1*	NM_003365	Ubiquinol-cytochrome c reductase core protein I
G09	*UQCRC2*	NM_003366	Ubiquinol-cytochrome c reductase core protein II
G10	*UQCRFS1*	NM_006003	Ubiquinol-cytochrome c reductase, Rieske iron-sulfur polypeptide 1
G11	*UQCRH*	NM_006004	Ubiquinol-cytochrome c reductase hinge protein
G12	*UQCRQ*	NM_014402	Ubiquinol-cytochrome c reductase, complex III subunit VII, 9.5 kDa
H01	*B2M*	NM_004048	Beta-2-microglobulin
H02	*HPRT1*	NM_000194	Hypoxanthine phosphoribosyltransferase 1
H03	*RPL13A*	NM_012423	Ribosomal protein L13a
H04	*GAPDH*	NM_002046	Glyceraldehyde-3-phosphate dehydrogenase
H05	*ACTB*	NM_001101	Actin, beta
H06	*HGDC*	SA_00105	Human Genomic DNA Contamination
H07	*RTC*	SA_00104	Reverse Transcription Control
H08	*RTC*	SA_00104	Reverse Transcription Control
H09	*RTC*	SA_00104	Reverse Transcription Control
H10	*PPC*	SA_00103	Positive PCR Control
H11	*PPC*	SA_00103	Positive PCR Control
H12	*PPC*	SA_00103	Positive PCR Control

**Table 5 ijms-24-15192-t005:** Panel of gene expression analyzed using the Human Mitochondria RT^2^ Profiler PCR Array. This array profiled the expression of 84 genes involved in diverse mitochondrial function. The transcripts monitored by this array encoded proteins which are regulators of mitochondrial biogenesis, regulators and mediators of mitochondrial molecular transport, and genes involved in apoptosis. Position indicates the location in the 96-well plate where the gene was assessed, Symbol denotes the gene identification, RefSeq denotes the Reference Sequence from the National Center for Biotechnology Information collection, and Description gives summary information about the gene identification and/or function.

Position	Symbol	Refseq	Description
A01	*AIFM2*	NM_032797	Apoptosis-inducing factor, mitochondrion-associated, 2
A02	*AIP*	NM_003977	Aryl hydrocarbon receptor-interacting protein
A03	*BAK1*	NM_001188	BCL2-antagonist/killer 1
A04	*BBC3*	NM_014417	BCL2 binding component 3
A05	*BCL2*	NM_000633	B-cell CLL/lymphoma 2, apoptosis regulator
A06	*BCL2L1*	NM_138578	BCL2-like 1, apoptosis regulator BCLX
A07	*BID*	NM_001196	BH3-interacting domain death agonist
A08	*BNIP3*	NM_004052	BCL2/adenovirus E1B 19kDa-interacting protein 3, pro-apoptotic factor
A09	*CDKN2A*	NM_000077	Cyclin-dependent kinase inhibitor 2A, inhibits CDK4
A10	*COX10*	NM_001303	COX10 cytochrome c oxidase assembly protein homolog
A11	*COX18*	NM_173827	COX18 cytochrome c oxidase assembly homolog
A12	*CPT1B*	NM_004377	Carnitine palmitoyltransferase 1B
B01	*CPT2*	NM_000098	Carnitine palmitoyltransferase 2
B02	*DNAJC19*	NM_145261	DnaJ (Hsp40) homolog, subfamily C, member 19, TIMM14
B03	*DNM1L*	NM_005690	Dynamin 1-like, mitochondrial and peroxisomal division
B04	*FIS1*	NM_016068	Mitochondrial fission 1 protein homolog
B05	*TIMM10B*	NM_012192	Translocase of inner mitochondrial membrane 10 homolog B
B06	*GRPEL1*	NM_025196	GrpE-like 1, mitochondrial protein import
B07	*HSP90AA1*	NM_001017963	Heat shock protein 90kDa alpha, class A member 1, folding of target proteins
B08	*HSPD1*	NM_002156	Heat shock 60kDa protein 1, chaperonin family, folding and assembly of proteins
B09	*IMMP1L*	NM_144981	Mitochondrial inner membrane protease subunit 1-like
B10	*IMMP2L*	NM_032549	Mitochondrial inner membrane protease Subunit 2-like
B11	*LRPPRC*	NM_133259	Leucine-rich PPR-motif containing, cytoskeletal organization and vesicular transport
B12	*MFN1*	NM_033540	Mitofusin 1, mediator of mitochondrial fusion
C01	*MFN2*	NM_014874	Mitofusin 2, mediator of mitochondrial fusion
C02	*MIPEP*	NM_005932	Mitochondrial intermediate peptidase, maturation of OXPHOS-related proteins
C03	*MSTO1*	NM_018116	Misato homolog 1, mitochondrial distribution and morphology regulator
C04	*MTX2*	NM_006554	Metaxin 2, mitochondrial outer membrane import complex protein 2
C05	*NEFL*	NM_006158	Neurofilament, light polypeptide, protein phosphatase 1
C06	*OPA1*	NM_130837	Optic atrophy 1, mitochondrial dynamin-like GTPase, related to mitochondrial network
C07	*PMAIP1*	NM_021127	Phorbol-12-myristate-13-acetate-induced protein 1, related to activation of caspases and apoptosis
C08	*RHOT1*	NM_018307	Ras homolog gene family, member T1, mitochondrial GTPase involved in mitochondrial trafficking
C09	*RHOT2*	NM_138769	Ras homolog gene family, member T2, mitochondrial GTPase involved in mitochondrial trafficking
C10	*SFN*	NM_006142	Stratifin
C11	*SH3GLB1*	NM_016009	SH3-domain GRB2-like endophilin B1, Bax-interacting Factor 1, apoptotic signaling pathway
C12	*SLC25A1*	NM_005984	Solute carrier family 25 (mitochondrial carrier; citrate transporter), member 1
D01	*SLC25A10*	NM_012140	Solute carrier family 25 (mitochondrial carrier; dicarboxylate transporter), member 10
D02	*SLC25A12*	NM_003705	Solute carrier Family 25 (aspartate/glutamate carrier), member 12, calcium carrier
D03	*SLC25A13*	NM_014251	Solute carrier Family 25 (aspartate/glutamate carrier), member 13
D04	*SLC25A14*	NM_003951	Solute carrier family 25 (mitochondrial carrier), member 14, UCP5
D05	*SLC25A15*	NM_014252	Solute carrier family 25 (mitochondrial carrier; ornithine transporter) member 15
D06	*SLC25A16*	NM_152707	Solute carrier family 25 (mitochondrial carrier), member 16
D07	*SLC25A17*	NM_006358	Solute carrier family 25 (mitochondrial carrier; peroxisomal membrane protein), member 17
D08	*SLC25A19*	NM_021734	Solute carrier family 25 (mitochondrial thiamine pyrophosphate carrier), member 19
D09	*SLC25A2*	NM_031947	Solute carrier family 25 (mitochondrial carrier; ornithine transporter) member 2, ORNT2
D10	*SLC25A20*	NM_000387	Solute carrier family 25 (carnitine/acylcarnitine translocase), member 20
D11	*SLC25A21*	NM_030631	Solute carrier family 25 (mitochondrial oxodicarboxylate carrier), member 21
D12	*SLC25A22*	NM_024698	Solute carrier family 25 (mitochondrial carrier: glutamate), member 22
E01	*SLC25A23*	NM_024103	Solute carrier family 25 (mitochondrial carrier; phosphate carrier), member 23
E02	*SLC25A24*	NM_013386	Solute carrier family 25 (mitochondrial carrier; phosphate carrier), member 24
E03	*SLC25A25*	NM_052901	Solute carrier family 25 (mitochondrial carrier; phosphate carrier), member 25
E04	*SLC25A27*	NM_004277	Solute carrier family 25, member 27, UCP4
E05	*SLC25A3*	NM_002635	Solute carrier family 25 (mitochondrial carrier; phosphate carrier), member 3
E06	*SLC25A30*	NM_001010875	Solute carrier family 25, member 30
E07	*SLC25A31*	NM_031291	Solute carrier family 25 (mitochondrial carrier; adenine nucleotide translocator), member 31, ANT4
E08	*SLC25A37*	NM_016612	Solute carrier family 25, (mitochondrial iron transporter), member 37
E09	*SLC25A4*	NM_001151	Solute carrier family 25 (mitochondrial carrier; adenine nucleotide translocator), member 4, ANT1
E10	*SLC25A5*	NM_001152	Solute carrier family 25 (mitochondrial carrier; adenine nucleotide translocator), member 5, ANT2
E11	*SOD1*	NM_000454	Superoxide dismutase 1, soluble, Cu/Zn superoxide dismutase
E12	*SOD2*	NM_000636	Superoxide dismutase 2, mitochondrial, Fe/Mn superoxide dismutase
F01	*STARD3*	NM_006804	StAR-related lipid transfer (START) domain containing 3, lipid trafficking protein
F02	*TAZ*	NM_000116	Tafazzin
F03	*TIMM10*	NM_012456	Translocase of inner mitochondrial membrane 10 homolog (yeast)
F04	*TIMM17A*	NM_006335	Translocase of inner mitochondrial membrane 17 homolog A (yeast)
F05	*TIMM17B*	NM_005834	Translocase of inner mitochondrial membrane 17 homolog B (yeast)
F06	*TIMM22*	NM_013337	Translocase of inner mitochondrial membrane 22 homolog (yeast)
F07	*TIMM23*	NM_006327	Translocase of inner mitochondrial membrane 23 homolog (yeast)
F08	*TIMM44*	NM_006351	Translocase of inner mitochondrial membrane 44 homolog (yeast)
F09	*TIMM50*	NM_001001563	Translocase of inner mitochondrial membrane 50 homolog (S. cerevisiae)
F10	*TIMM8A*	NM_004085	Translocase of inner mitochondrial membrane 8 homolog A (yeast)
F11	*TIMM8B*	NM_012459	Translocase of inner mitochondrial membrane 8 homolog B (yeast)
F12	*TIMM9*	NM_012460	Translocase of inner mitochondrial membrane 9 homolog (yeast)
G01	*TOMM20*	NM_014765	Translocase of outer mitochondrial membrane 20 homolog (yeast)
G02	*TOMM22*	NM_020243	Translocase of outer mitochondrial membrane 22 homolog (yeast)
G03	*TOMM34*	NM_006809	Translocase of outer mitochondrial membrane 34
G04	*TOMM40*	NM_006114	Translocase of outer mitochondrial membrane 40 homolog (yeast)
G05	*TOMM40L*	NM_032174	Translocase of outer mitochondrial membrane 40 homolog (yeast)-like
G06	*TOMM70A*	NM_014820	Translocase of outer mitochondrial membrane 70 homolog A (S. cerevisiae)
G07	*TP53*	NM_000546	Tumor protein p53, P53 tumor suppressor
G08	*TSPO*	NM_000714	Translocator protein (18kDa), transport of cholesterol
G09	*UCP1*	NM_021833	Uncoupling protein 1 (mitochondrial, proton carrier), SLC25A7, proton leak
G10	*UCP2*	NM_003355	Uncoupling protein 2 (mitochondrial, proton carrier), SLC25A8, proton leak
G11	*UCP3*	NM_003356	Uncoupling protein 3 (mitochondrial, proton carrier), SLC25A9, proton leak
G12	*UXT*	NM_004182	Ubiquitously expressed transcript
H01	*B2M*	NM_004048	Beta-2-microglobulin
H02	*HPRT1*	NM_000194	Hypoxanthine phosphoribosyltransferase 1
H03	*RPL13A*	NM_012423	Ribosomal protein L13a
H04	*GAPDH*	NM_002046	Glyceraldehyde-3-phosphate dehydrogenase
H05	*ACTB*	NM_001101	Actin, beta
H06	*HGDC*	SA_00105	Human Genomic DNA Contamination
H07	*RTC*	SA_00104	Reverse Transcription Control
H08	*RTC*	SA_00104	Reverse Transcription Control
H09	*RTC*	SA_00104	Reverse Transcription Control
H10	*PPC*	SA_00103	Positive PCR Control
H11	*PPC*	SA_00103	Positive PCR Control
H12	*PPC*	SA_00103	Positive PCR Control

## Data Availability

The data presented in this study are available on request from the corresponding author.

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
