# Peer review of "Characterizing Early Cardiac Metabolic Programming via 30% Maternal Nutrient Reduction during Fetal Development in a Non-Human Primate Model"

_ijms, 2023, doi:10.3390/ijms242015192_

Round 1
Reviewer 1 Report
In this manuscript, the authors sought to understand the effect of dietary restrictions on fetal growth and heart function by using non-human primate models. The experiments are well-designed. The results show that caloric restriction modulates genes and protein expression in the fetal heart. I also appreciate the authors provided male and female differences. The results are very intriguing and will provide novel aspects regarding the nutritional regulation of heart function in primates. There are a few points that need to be addressed before publication.
Major points:
1) In Fig. 6, the authors showed Western blot images and quantification. The overall images and sample group labeling are not clear. The authors must provide clear and larger images of the blots to support the quantification.
2) Mitochondrial functional assay is missing. Although the authors showed the changes in genes and proteins important for mitochondrial function, the function of mitochondria is unknown. Therefore, I suggest the authors perform mitochondrial OxPhos complex activity and Citrate synthase activity assays using heart tissues.
3) The mechanism is not clear. In the discussion section, the authors need to provide mechanistic insight into the nutrition-induced alteration of mitochondrial genes and protein changes. Why does the reduction of nutrients affect fetal gene expression for mitochondrial function? Which pathway is affected by nutrition?
Reviewer 2 Report
The Authors have carried out a well designed and implemented animal study, with a simple but yet mostly effective investigation ex-vivo.
The premise of the study is that reduced nutrition during pregnancy results in altered organ (cardiac) function as a result of mitochondrial dysregulation. The animal model is sound, but the analysis somewhat simplistic.
If the authors wanted to evaluate mitochondrial function, imaging and preferably seahorse assays could have been utilised. There are also imaging technologies which allow mitochondrial enumeration with simultaneous ROS measurements in frozen sections. If the authors have such data I would encourage them to include it. I think that it's a pity that the authors only used PCR and WB to analyse the data. Adding in other types of analysis would enhance the robustness of the study, but I doubt would change the conclusions, and therefore are additive but not necessary.
The introduction alludes to "programming", but no description of how this could occur with mitochondria. I would guess that the authors are aware that a large percentage of mitochondrial proteins are encoded genetically and are susceptible to epigenetic programming, perhaps this could be included somewhere.
The tables are a little confusing with data in the maternal group being entered under "males". Was this the fathers? Did the fathers really get heavier with dietary restriction?
Do you have data on mitochondria in other organs at this time point - if you do it would be interesting to know if the excepts observed are seen elsewhere?
Could you help me to understand the WB data. Were mitochondria extracted or is this whole cell lysate? if the latter why wasn't a typical house keeping gene used. Was equal quantities of protein loaded? The images of the gels do not help in the PDF that I am viewing.
Round 2
Reviewer 1 Report
The authors answered all questions.